# Antihypertensive Effect of Galegine from *Biebersteinia heterostemon* in Rats

**DOI:** 10.3390/molecules26164830

**Published:** 2021-08-11

**Authors:** Weien Wang, Xiaofeng Zhang

**Affiliations:** 1School of Chemistry and Chemical Engineering, Qinghai Normal University, Xining 810008, China; 2Northwest Institute of Plateau Biology, Chinese Academy of Sciences, Xining 810007, China; xfzhang@126.com; 3School of Chemistry, University of Chinese Academy of Sciences, Beijing 100049, China

**Keywords:** *Biebersteinia heterostemon*, galegine, hypotensive, toxicity

## Abstract

The aerial part of *Biebersteinia heterostemon* Maxim. (Geraniaceae Biebersteiniaceae) known as *ming jian na bao* in Chinese, has been traditionally used in Tibetan folk medicine for treatment of diabetes and hypertension. The aim of the present study was to evaluate the effects of galegine obtained from an ethanol extract of the entire *Biebersteinia heterostemon* plant on the rat’s cardiovascular system in order to characterize its contributions as an antihypertensive agent. The antihypertensive effect of galegine was investigated in pentobarbital-anesthetized hypertensive rats at three dose levels based on the LD_50_ of galegine. Meanwhile a positive control group received dimaprit with the same procedure. Dimaprit infusion induced a significant hypotension which declined by an average margin of 20%. Simultaneously, single administration of galegine at the doses of 2.5, 5, and 10 mg/kg by intraperitoneal injection induced an immediate and dose-dependent decrease in mean arterial blood pressure (MABP) by an average margin of 40% with a rapid increase in heart rate (HR). We demonstrated that galegine is effective in reducing blood pressure in anesthetized hypertensive rats with rapid onset and a dose-related duration of the effects. The results indicate that galegine was the bioactive compound which can be used as a pharmacophore to design new hypertensive agents.

## 1. Introduction

Hypertension is a major risk factor for stroke, myocardial infarction, heart failure, and kidney failure. Worldwide, hypertension is estimated to cause 9.4 million premature deaths and counts for 4.5% of disease [1,2]. Treatment of hypertension is associated with a reduction in the risk of stroke of approximately 40% and a reduction in the risk of myocardial infarction of approximately 15% [3]. Consequently, guidelines on clinical practice have identified lowering of blood pressure (BP) as a priority in hypertension treatment [4]. However, hypertension can be managed in a suboptimal manner in many countries.

Natural products have made many unique and vital contributions to drug discovery. Several hypertensive agents have been derived from pharmacophores (i.e., a part of a molecular structure responsible for a particular biologic/pharmacologic interaction that it undergoes) from natural products. The treatment of hypertension with plant-derived products is well known, such as (+)-Dicentrine, Rhynchophylline, Stevioside, ACE inhibitory peptides and so on [5]. The potential value of herbal medicines for hypertension treatment has been rediscovered [6]. Therefore, pharmacologic validation of medicinal plants or ethnomedical treatment methods could benefit development of new drugs.



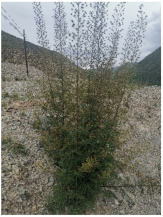



The aerial part of *Biebersteinia heterostemon* Maxim. (Geraniaceae), known as *ming jian na bao* in Chinese, has been used in Tibetan folk medicine for treatment of diarrhea, edema, apoplexy, stomach pain, anthrax, erysipelas, and malaria. In Qinghai (Tibet, China), the entire plant is used by traditional healers for treatment of diabetes mellitus and hypertension [7,8]. However, preparation of *B. heterostemon* as an antihypertensive agent has not been described in detail. Only phytochemical studies carried out with the entire *B. heterostemon* plant, which have led to the isolation of guanidine alkaloids comprising mainly galegine and 4-hydroxygalegine [9], have been mentioned. Galegine has been reported to cause hyperglycemia if isolated as an alkaloid from *Verbesina encelioides* and *Galega officinalis* (which contains 0.1–0.3% galegine). Study of the hypoglycemic properties of galegine led to the discovery of metformin [10,11]. Studies have also shown that galegine can reduce weight indirectly by inhibiting the synthesis and stimulating the oxidation of fatty acids [12]. However, *B. heterostemon* has not been studied specifically for its cardiovascular effects or mechanism of action of its antihypertensive effects. Therefore, the aim of the present study was to evaluate the effects of galegine obtained from an ethanol extract of the entire *B. heterostemon* plant on the cardiovascular system of rats so that its contributions as an antihypertensive agent could be characterized.

## 2. Materials and Methods

### 2.1. Isolation of Plant Material

The aerial parts of *B. heterostemon* in the blooming phase were collected in Tongren County (Qinghai province, Tibet, China) in August 2014. These aerial parts were identified by Professor Xuefeng Lu (Department of Botany, Northwest Institute of Plateau Biology, Chinese Academy of Sciences, Qinghai Sheng, China). A voucher specimen (number 98,018) was deposited at the Herbarium of Tibetan Medicinal Plants (0028, holotypus) at the Northwest Institute of Plateau Biology.

Air-dried and finely ground aerial parts (20 kg) of *B. heterostemon* were extracted three times, once every 5 days, with 20 L 90% EtOH at room temperature. The concentrated syrup was suspended in H_2_O then partitioned successively with petroleum ether, AcOEt, and n-BuOH, with a residue yield of 5 g, 138 g, and 460 g, respectively, after solvent removal. As part of a search for antihypertensive principles by bioassay-directed separation, the AcOEt extract was named GAP, which was selected for study because previous phytochemical studies have revealed that GAP contains galegine.

Half of the GAP (50 g) was chromatographed over a normal silica-gel column (40–63 μm, 5 × 120 cm) eluted with solvents of increasing polarity in the order petrol-AcOEt (10:1–2:1), petrol-acetone (10:1–10:3), CHCl_3_-acetone (10:1–1:1), and acetone. Chromatography was monitored by thin-layer chromatography (petrol-AcOEt, 1:1; CHCl_3_-acetone, 4:1). Fractions eluted with CHCl_3_-acetone 10:3 (following elution with 3 L of this solvent, and fractions of 250 mL were collected, and were collectively named PCF according to the elution order) gave compound **4** (990 mg), which was purified by column chromatography and recrystallization.

The content of galegine in the GAP extract is 1.98% and the participation of galegine is a pure compound in the pharmacological effect.

### 2.2. Animals

The experimental protocol for animal studies was approved by local Animal Ethics Committees in accordance with the guidelines for the care and use of laboratory animals set by the Faculty of Medicine of Qinghai University (Xining, China), and incompliance with national (GB/T 35892-2018) and international rules on care and use of laboratory animals (NIH Publication No. 85-23, rewised by 1985). All tests were performed during the light phase.

Male Sprague Dawley rats (4 weeks, 90 ± 6 g) and Kunming rats of both sexes (10–12 weeks, 25–30 g) were purchased from the Institute of Local Disease (Xining, China). Rats were kept in a room under automatically controlled conditions of 22 ± 1 °C and a 12-h light–dark cycle. Rats were fed standard laboratory diet provided by the Institute of Local Disease and were allowed to acclimatize to their surroundings for ≥1 week before experimentation.

### 2.3. Acute Toxicity

Kunming rats of both sexes (10–12 weeks, 25–30 g) were used for acute toxicity studies. Rats were divided into five groups with graded randomization to make the mean weight and sex distribution as equal as possible. Each group comprised 5 males and 5 females. An acute study for calculation of the median lethal dose (LD_50_) was carried out using Karber’s method as modified by Sun and colleagues [13]. In this method, the galegine dose was determined through pre-testing. Rats were fasted overnight before conducting the experiment but had free access to water. The diluted drug was injected by intraperitoneal injection (i.p.) taking 1.25 as the geometric proportion between groups to administrate the dosage volume of galegine by base dosage (80 mg/kg. body weight) in each group.

Detailed clinical observations were made for all rats throughout the study. Body weights were recorded on the day of treatment and on test days 4, 7, 10, 13, and 16. Every 24 h, the dose for each group and the number of dead rats were recorded. Necropsies were carried out as soon as possible after death on all rats that died during the study. At the end of the study, all surviving rats were sacrificed.

For acute toxicity, we used death rates in the group with a minimum dose of 0% and the group with a maximum dose of 100%. LD_50_ values and 95% confidence limits (A) were calculated as follows:(1)$${\mathrm{LD}}_{50}={\mathrm{lg}}^{-1}\left[X_{m}-i\left(\sum p-0.5\right)\right]$$
(2)$$A={\mathrm{lg}}^{-1}\left({\mathrm{lgLD}}_{50}\pm 1.96\times i\times \sqrt{\frac{\sum p\left(1-p\right)}{n}}\right)$$
where *X_m_* is the logarithm of maximum dose; *i* represents the logarithm difference between two adjacent doses; Σ*p* is the sum of the mortality of animals; *n* is number of rats per group.

Galegine was administered to five groups in doses of 32.8, 40.96, 51.5, 64, and 80 mg/kg body weight, and the death rate in each group was 0, 10%, 40%, 70%, and 100%, respectively.

### 2.4. Feeding of a Refined-Sugar Preparation to Rats

Studies were carried out in male Sprague Dawley (4 weeks, 90 ± 6 g). Rats were separated into two groups: I (control rats given tap water) and II (rats given 30% of commercially refined sugar in drinking water). These rats were given the respective drinks for 16 weeks. Sugar treatment ended immediately after the mean BP of group-II rats increased significantly from 90–110 mmHg to 120–150 mmHg. This model is characterized by hyperinsulinemia, loss of tissue sensitivity towards insulin, hypertriglyceridemia, arterial hypertension, and an increase in oxidative stress [14]. Systolic arterial pressure (SAP) measurements were taken every month. Rats were maintained at 32 °C in a LE 5650/6 heater and scanner heating unit (Letica, Rochester Hills, MI, USA). A pulse transducer and pressure cuff (LE 5160/R, pulse transducer and pressure cuff for rats, Harvard Appratus, Holliston, MA, USA) were placed around the tail of each rat and connected to an automatic blood-pressure system (LE 5007, RCA connector, Ningbo Hysound Electronic Co., Ltd., Ningbo, China). After 16 weeks, we selected rats whose SAP had increased ≤30% to be the hypertensive rats from group II.

### 2.5. Experimental Protocol

Six male hypertensive Sprague Dawley rats (20 weeks, 280–320 g) were anesthetized successively (pentobarbital sodium, 40 mg/kg body weight, i.p.). The common carotid artery was cannulated and BP monitored using a pressure transducer (BL-410 biological experimental system; Sichuan Tai Meng Technology, Chengdu, China), which was triggered by the pressure pulse and recorded on a separate polygraph channel. Various concentrations of galegine were injected (i.p.) into rats, and the effects on arterial blood pressure (ABP) and heart rate (HR) recorded. Each rat was tested with only one concentration. ABP was expressed as the mean arterial blood pressure (MABP) according to the following equation:(3)$$\mathrm{MABP}=P_{d}+\frac{1}{3}\left(P_{s}-P_{d}\right)\;\mathrm{mmHg}$$
where *P*_s_ denote systolic BP and *P*_d_ denotes diastolic BP.

The common carotid artery was excised rapidly. Then, the distal part of the heart was ligated with thread tightly. The proximal part of the heart was clamped by artery forceps. A V-shaped incision was made near the ligation point. An arterial cannula filled with 0.1% heparinized saline was inserted. The arterial cannula was connected to the force–displacement transducer linked to the physiologic-pressure detector. After the pressure detector had been adjusted, the artery forceps were opened and normal BP recorded. After a stabilization period of 15 min, single administration of galegine (2.5, 5 or 10 mg/kg) or saline solution (0.2 mL/100 g) was injected (i.p.) in six experimental rats. An additional positive control group received dimaprit (5 mg/kg body weight, 99% purity, Sigma–Aldrich, St. Louis, MO, USA) under the same procedure. In this series, recordings of MABP and HR were taken immediately over 60 min and their values were registered every 5 min. In sum, each group contained 6 rats and their MABP and HR were recorded 12 times over 60 min (before and every 5 min). We obtained six values for each of those 12 recordings, from which we present a median. The male Sprague Dawley rats were sacrificed after the experiment.

### 2.6. Statistical Analyses

Data are the mean ± SEM. Student’s *t*-test, one-way analysis of variance (ANOVA), and post hoc least-significant difference tests were used to determine significant differences between groups. *p* < 0.05 was considered significant.

## 3. Results

The structure was demonstrated to be galegine (compound **4**, Figure 1) by two-dimensional nuclear magnetic resonance (NMR) and high-resolution mass spectrometry.

GAP was subjected to reversed-phase column chromatography (C_18_, 5 μm, 250 mm × 4.6 mm i.d.; Thermo Fisher Scientific, Waltham, MA, USA) to high-performance liquid chromatography with diode-array detection (HPLC–DAD) analyses. Spectrometric analyses were carried out with a HPLC system (Waters, Milford, MA, USA) comprising a 1525 binary pump and 2996 photodiode array detector.

PCF (20 mg) was dissolved in 1 mL of H_2_O–MeOH (80:20) and injected into the C_18_ cartridge. Then, 2 mL of H_2_O–MeOH 80:20 (*v*/*v*) was applied to the cartridge for rinsing. The achieved retained sample was eluted with a mixture of 2 mL H_2_O–MeOH 50:50 (*v*/*v*). This mixture displaced GAP and showed an intense narrow ring proceeding downwards, which was monitored by the naked eye. Parameters for this process were: flow rate = 1 mL/min; injection volume = 20 μL; concentration of galegine sample = 10 mg/mL in H_2_O–MeOH 50:50 (*v*/*v*); DAD conditions = 205 nm. The HPLC profile of GAP and galegine (peak 1) is shown in Figure 2.

Compound **4** was obtained as colorless, needle-like crystals. m.p. 104~105 °C. A positive Sakaguchi reaction suggested that this compound could be a guanidine alkaloid. High-resolution electrospray ionization mass spectrometry exhibited a molecular ion peak [M + H]^+^ at *m*/*z* 128, which corresponded to the molecular formula C_6_H_13_N_3_. Infrared absorption bands at 3405, 3201, and 1676 cm^−1^ suggested the presence of primary amines and secondary amines. According to ^1^H and ^13^C NMR data (Table 1), the structure of compound **4** was determined to be galegine. These data were identical to those of galegine [9].

The galegine (purity = 99.2%; 990 mg) used in the experiments was prepared fresh by dissolving in distilled water.

### 3.1. Lethality and Clinical Signs

Galegine was administered by intraperitoneal injection (IP) to five groups of fifty mice at the doses of 32.8, 40.96, 51.5, 64, and 80 mg kg^−1^, and we recorded the clinical signs and calculated the toxin median lethal dose (LD_50_), based on 24 h lethality data.

At the dose of 41 mg kg^−1^ and above, galegine administration provoked an onset of clinical signs (prostration, tremors, followed by abdominal breathing, paralysis of the hindlimbs, and cyanosis), which led to the death of mice within less than 18 h. In particular, the lowest lethal dose (41 mg kg^−1^) provoked the death of 1/10 mice, while 80 mg kg^−1^ was lethal for 10/10 mice (Table 2). These results are presented in Figure 3 as percentage of mice mortality versus the administered toxin doses. Based on lethality data, the oral LD_50_ of galegine was calculated at 54.75 mg kg^−1^ (95% confidence limits: 49.15–61.51 mg kg^−1^).

### 3.2. Effect of Galegine on Arterial BP

To evaluate the acute hypotensive effects of dimaprit and galegine in vivo, arterial blood pressure (ABP) was continuously measured through a pressure transducer inserted in the carotid artery of anesthetized rats. Intraperitoneal injection (i.p.) of dimaprit and galegine significantly decreased ABP and MABP, as expected (Figure 4A,B). To assess the intoxication protocol, we designed a negative control experiment (control group) where rats received an injection of distilled water: no significant change in ABP was observed (Figure 5, Table 3). Intraperitoneal injection (i.p.) of dimaprit was designed as THE positive control group.

A dose of 5 mg of dimaprit infusion induced significant hypotension (MABP declined by an average of 20%) and was associated with the slight increase of HR. The blood pressure decreased and was maintained for 15 min. Simultaneously, injection of galegine (2.5, 5, and 10 mg·kg^−1^, i.p.) induced an immediate and dose-dependent decrease in MABP by an average of 36%, 40%, and 43% respectively (Figure 4, Table 3) with a rapid increase in HR (Table 3). The blood pressure decreased and was maintained for 28, 32, and 45 min, respectively. In contrast, at the same dose, the hypotensive effect of galegine was better and lasted longer. However, after this peak in hypotension, the MABP increased progressively and reached the initial basal value in approximately 10–15 min depending on galegine dose.

## 4. Discussions

The guanidine functional group has been reported to possess hypotensive properties. However, few investigations have been done on the hypotensive effects of galegine, an archetypal guanidine alkaloid. Here, we demonstrated that galegine is effective in reducing ABP in anesthetized hypertensive rats with a rapid onset and dose-related duration of effects using a pressure transducer to record ABP. The safe dose of galegine was determined to be one-tenth of the LD_50_ value after acute toxicity experiments. Galegine showed physiological effects such as vasodilatation and hypotension [15]. A dose of 2.5 mg of galegine appears to be effective in managing hypertensivity, and the hypotensive effect of galegine was better and lasted longer. This indicated that galegine could be effective in managing hypertensive urgency and controlling blood pressure [5]. Although the antihypertensive mechanisms were not clear, it has been reported that the hypotensive mechanism of galegine is related to the H_2_-receptor agonist [16]. Thus, a highly selective agonist of the histamine H_2_ receptor, dimaprit, could mimic the excitatory effect of histamine on rubral cells, and the histaminergic nervous system may have a modulatory role in motor control through its excitatory effects on almost all subcortical motor structures [17,18,19]. Several ionic channels and intracellular signaling pathways have also been suggested to mediate the excitatory response of central neurons to histamine stimulation [20]. Thus, further studies, such as drug receptor-specific interaction tests or chronic toxicity tests, should be carried out to confirm the long-term safety of galegine for BP control.

## Figures and Tables

**Figure 1 molecules-26-04830-f001:**
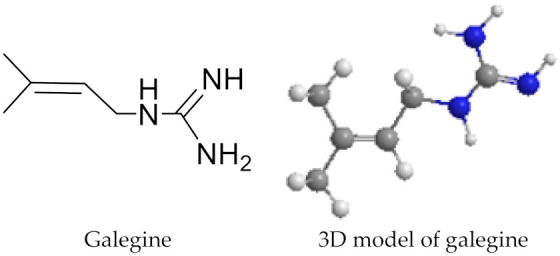
Scheme of the structure of galegine.

**Figure 2 molecules-26-04830-f002:**
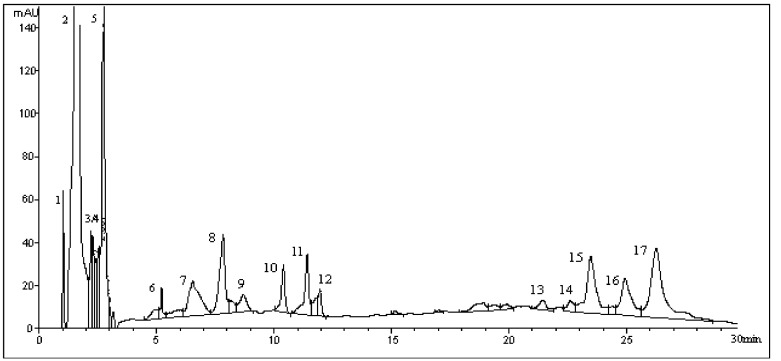
HPLC profile recorded at 255 nm. *Biebersteinia heterostemon* bark antihypertensive fraction (GAP).

**Figure 3 molecules-26-04830-f003:**
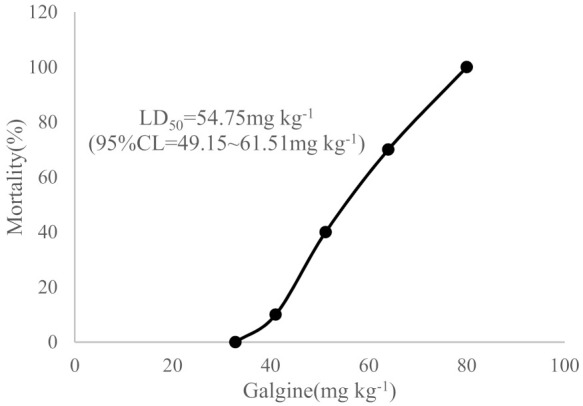
Dose-response mortality curve of galegine after i.p. injections administration in mice. Percentage lethality is plotted against the administered doses of galegine.

**Figure 4 molecules-26-04830-f004:**
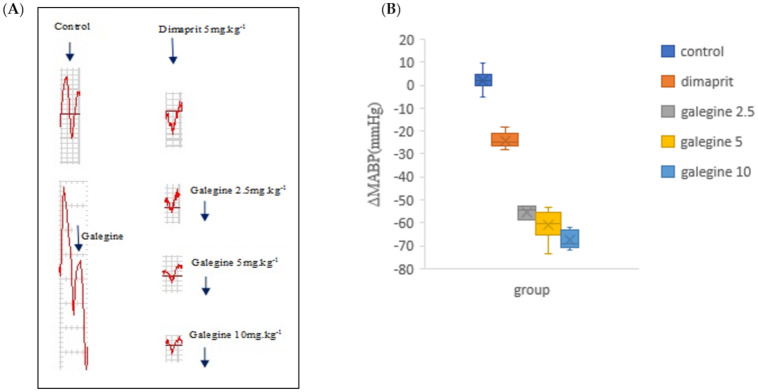
Influence of dimaprit and galegine on pentobarbital-anesthetized hypertensive rats MABP. (**A**) Representative raw traces of rat arterial blood pressure (ABP) following dimaprit or galegine infusion (blue arrow with doses used) compared to the control group. As expected, protocol assessment molecules dimaprit (5 mg·kg^−^^1^) and galegine (2.5 mg·kg^−1^, 5 mg·kg^−1^, 10 mg·kg^−1^) induced ABP, respectively. (**B**) Box and whisker plots of ΔMABP data in the control, dimaprit and galegine group following i.p. injections, 5 mg·kg^−1^ for dimaprit and 2.5 mg·kg^−1^, 5 mg·kg^−1^, 10 mg·kg^−1^ for galegine, respectively.

**Figure 5 molecules-26-04830-f005:**
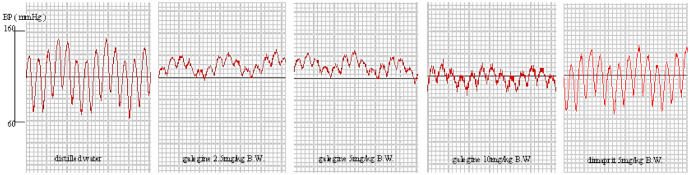
Effect of galegine on blood pressure of pentobarbital-anesthetized hypertensive rats. *n* = 6.

**Table 1 molecules-26-04830-t001:** ^1^H and ^13^C NMR spectral data of galegine.

	δH ^a^	δC ^b^
1(-CH2)	3.69 (d, 6.4)	40.1
2(-CH)	5.17 (*br t*, 6.0)	118.9
3		136.3
4(-CH_3_)	1.69 *br s*	25.2
3(-CH_3_)	1.63 *br s*	17.7
C=NH	7.50 *br s*	156.5

Measured in DMSO-d_6_. ^a^ 400 MH_Z_. ^b^ 150 MH_Z_. Data are expressed as mean ± S.E.M. (*n* = 6). Significantly different between before and after treatment.

**Table 2 molecules-26-04830-t002:** Lethality and signs of toxicity of mice after intraperitoneal injection administration of galegine.

Group of Treatment	Dose (mg·kg^−1^)	Lethality	Survival Times (hour)	Signs of Toxicity
Controls	-	0/10	-	
Galgine	32.8	0/10	-	-
	41	1/10	18	Debility, abdominal breathing, paralysis of the hindlimbs, cyanosis
	51.2	4/10	9.5–10–10.5–10.5	Prostration, tremors, abdominal breathing, paralysis of the hindlimbs, cyanosis
	64	7/10	9–9–9.5–9.5–9.6–9.7–9.7	Prostration, tremors, abdominal breathing, paralysis of the hindlimbs, cyanosis
	80	10/10	6–6–7-7–7–7.5–7.6–7.6–8–8	Prostration, tremors, abdominal breathing, jumping, paralysis of the hindlimbs, cyanosis

**Table 3 molecules-26-04830-t003:** Effect of galegine on mean arterial blood pressure (MABP) of pentobarbital-anesthetized hypertensive rats.

Group	MABP (mmHg)		Decrease of MABP (%)	Heart Rate (beats/min)	Duration of Hypotension Period (min)
	Before Treatment	After Treatment			
Control	98.25 ± 3.22	100.65 ± 4.32	0.66 ± 0.07	427 ± 20	0.15 ± 0.16
Dimaprit 5 mg/kg	153.29 ± 5.21	122.62 ± 9.65	19.99 ± 0.02	457 ± 11	15.60 ± 0.20
Galegine 2.5 mg/kg	152.83 ± 7.03	96.83 ± 10.38	36.64 ± 0.05	602 ± 15	28.30 ± 0.20
Galegine 5 mg/kg	159.07 ± 4.92	93.08 ± 13.91	40.42 ± 0.08	850 ± 12	32.20 ± 0.16
Galegine 10 mg/kg	153.58 ± 9.13	86.60 ± 4.40	43.56 ± 0.02	1150 ± 20	45.50 ± 0.15

## Data Availability

All the data are contained within the article.

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
