# Peer review of "Antihypertensive Effect of Galegine from Biebersteinia heterostemon in Rats"

_molecules, 2021, doi:10.3390/molecules26164830_

Round 1
Reviewer 1 Report
The authors reported results from their structural study of Biebersteinia heteroestemon constituents and antihypertensive effect of galegine in rats among them. Although this study seems to be valuable as a result of a natural product chemistry study, the following points should be considered.
1) Since potent toxicity of galegine has been reported already, the authors should mention the novelty of this study more clearly.
2) The authors should give identity of the constituents other than galegine, corresponding to peak numbers of 1 – 3, and 5 – 17 shown in Figure 2.
3) The authors should show the content of galegine in the GAP extract, and the participation of galegine in the pharmacological effect of the extract.
Minor problems:
4) The plant family of the Biebersteinia plant species must be Biebersteiniaceae (not Geraniaceae) in the recent APG plant taxonomc study.
5) The expression “antihypotensive” (Line 5 in the abstract) must be “antihypertensive.”
6) Melting point of “needle-like crystals” (page 5) should be given.
7) The coupling constant of the triplet of C2-H should be given in Table 1.
Author Response
- Since potent toxicity of galegine has been reported already, the authors should mention the novelty of this study more clearly.
A dose of 5mg of dimaprit infusion induced significant hypotension (MABP declined by an average of 20%) and was associated with the slight increasement of HR. The blood pressure decreased and maintained for 15 minutes. Simultaneously, injection of galegine (2.5, 5, and 10 mg/kg, i.p.) induced an immediate and dose-dependent decrease in MABP by an average of 36%, 40% and 43% respectively (Figure 4, Table 3) with a rapid increase in HR (Table 3). The blood pressure decreased and maintained for 28,32 and 45 minutes respectively.
In contrast, at the same dose, the hypotensive effect of galegine was better and lasted longer. Those indicated that galegine could be effective in managing hypertensive urgency and to control the blood pressure
- The authors should give identity of the constituents other than galegine, corresponding to peak numbers of 1 – 3, and 5 – 17 shown in Figure 2.
Peak 1 is galegine in Figure 2, but corresponding to peak numbers of 2 – 3, and 5 – 17 haven’t enough data to give those identity.
3) The authors should show the content of galegine in the GAP extract, and the participation of galegine in the pharmacological effect of the extract.
The content of galegine in the GAP extract is 1.98%, and the participation of galegine is pure compound in the pharmacological effect .
Minor problems:
4) The plant family of the Biebersteinia plant species must be Biebersteiniaceae (not Geraniaceae) in the recent APG plant taxonomc study.
The wrong plant family of the Biebersteinia plant species has been corrected in paper and showed in red.
5) The expression “antihypotensive” (Line 5 in the abstract) must be “antihypertensive.”
The wrong expression “antihypotensive” (Line 5 in the abstract) has been corrected in paper and showed in red.
6) Melting point of “needle-like crystals” (page 5) should be given.
Melting point of “needle-like crystals” (page 5) has been given in paper and showed in red..
7) The coupling constant of the triplet of C2-H should be given in Table 1.
The coupling constant of the triplet of C2-H has been given in Table 1 and showed in red.

Reviewer 2 Report
The article “Antihypertensive effect of galegine from Biebersteinia heterostemon in rats” by Wang and Zhang has performed the extraction and purification of galegine from B heterostemon. The galegine acute toxicity is evaluated with its i.p LD50 on rats. Then galegine is challenged with its hypotensive activity on anesthetized rats.
This paper is interesting but should be significantly modified. It suffers from a lack of references (only 17 in the bibliography) and the discussion is weak. All the Discussion should be revised to really discuss the data presented. It’s not the case as it is.
This article should be proofread and corrected by a native English speaker as it contains too many errors.
The plant Biebersteinia heterostemon should be shown as a picture in the Introduction.
Introduction: « Several hypertensive agents have been derived from pharmacophores (i.e., a part of a molecular structure responsible for a particular biologic/pharmacologic interaction that it undergoes) from natural products”. You should give several examples and references of such natural products. Captopril probably…
Mat & Meth: no Institutional Review Board Statement for animal experiment has been added. No number or reference for animal experiment approval is indicated.
For Acute toxicity assay : I ask the authors to add a table and a figure. The table should describe the doses used, the symptoms observed for each dose, end the number of animals who died for each dose. The figure should show the lethality (%) as a function of the dose used. And the LD50 sould appear on the curve. See for example Table 1 and Figure 1 of https://doi.org/10.3390/toxins12020087
Acute toxicity: the LD50 is an acute assay done with a single exposure to galegine. Therefore, LD50 = 54.75 mg/kg. It is not 54.75 mg/kg body weight/day.
The section “Acute toxicity” should not contain the data on the anti-hypertensive action of galegine. There should be a section entitled “Effect of galegine on arterial BP”.
Figure 3: the time scale does not appear on the X axis. The BP variation should appear as mm Hg variation of DBP, not as a variation of BP values. See figure 5 of https://doi.org/10.3390/ijms22105106, which clearly shows hypertensive or hypotensive effects.
Discussion: “Galegine showed physiological effects such as vasodilatation”: please add a reference showing this effect.
Table 2:
- Where was measured the MABP (from Figure 3) ?
- Precise the’ statistical test used and to what * refers
- The HR values should be compared, since they drastically increase with galegine, suggesting a tachycardiac effect
- If the hypotension duration is 45.50 min long, this should appear on a graph
Minor
Abstract: “The antihypertensive effect of galegine was investigated” or The hypotensive effect of galegine was investigated”
Abstract: “Meanwhile a positive control group”
Abstract: “intraperitoneal injection of galegine”
Abstract: “at the doses of 2.5, 5, and 10 mg/kg”
Abstract: “was the main effective compound”
Introduction: myocardial infarction, heart and kidney failure
The 3D model of galegine is pixelized and should have a better resolution
Author Response
This paper is interesting but should be significantly modified. It suffers from a lack of references (only 17 in the bibliography) and the discussion is weak. All the Discussion should be revised to really discuss the data presented. It’s not the case as it is.
We have added references to 20 in the bibliography and enhanced the discussion.
This article should be proofread and corrected by a native English speaker as it contains too many errors.
I have polished the article for many times, and I hope get help for changing all erros.
The plant Biebersteinia heterostemon should be shown as a picture in the Introduction.
We show a picture of the plant Biebersteinia heterostemon in the Introduction
Introduction: « Several hypertensive agents have been derived from pharmacophores (i.e., a part of a molecular structure responsible for a particular biologic/pharmacologic interaction that it undergoes) from natural products”. You should give several examples and references of such natural products. Captopril probably…
We have given several examples and references of such natural products.
Mat & Meth: no Institutional Review Board Statement for animal experiment has been added. No number or reference for animal experiment approval is indicated.
The experimental protocol for animal studies was approved by local Animal Ethics Committees in accordance with the guidelines for the care and use of laboratory animals set by the Faculty of Medicine of Qinghai University (Qinghai, China), and incompliance with national (GB/T 35892-2018) and international rules on care and use of laboratory animals (NIH Publication No. 85-23, rewised by 1985). All test were performed during the light phase.
For Acute toxicity assay : I ask the authors to add a table and a figure. The table should describe the doses used, the symptoms observed for each dose, end the number of animals who died for each dose. The figure should show the lethality (%) as a function of the dose used. And the LD50 sould appear on the curve. See for example Table 1 and Figure 1 of https://doi.org/10.3390/toxins12020087
Yes, It’s a good suggestion, and we have done.
Acute toxicity: the LD50 is an acute assay done with a single exposure to galegine. Therefore, LD50 = 54.75 mg/kg. It is not 54.75 mg/kg body weight/day.
We corrected as LD50 = 54.75 mg/kg.
The section “Acute toxicity” should not contain the data on the anti-hypertensive action of galegine. There should be a section entitled “Effect of galegine on arterial BP”.
The section “Acute toxicity” has been corrected in paper and showed in red.
Figure 3: the time scale does not appear on the X axis. The BP variation should appear as mm Hg variation of DBP, not as a variation of BP values. See figure 5 of https://doi.org/10.3390/ijms22105106, which clearly shows hypertensive or hypotensive effects.
Now in Figure 4: Because we just cut a part of the picture from recorder, so there is no time scale on the X axis. In addition, we just show the range of BP, not as a variation of BP values. Do I clear myself ?
Discussion: “Galegine showed physiological effects such as vasodilatation”: please add a reference showing this effect.
Yes, we have added a reference showing this effect.
Table 2:
- Where was measured the MABP (from Figure 3) ?
- Precise the’ statistical test used and to what * refers
- The HR values should be compared, since they drastically increase with galegine, suggesting a tachycardiac effect
- If the hypotension duration is 45.50 min long, this should appear on a graph
Now in table3: MABP obtained from MABP = Pd + (Ps – Pd) mmHg, and Pd, Ps came from recorder(from Figure 4). In addition, the hypotension duration is 45.50 min long, which
Obtained from recorder. It was too long to appear on a graph.
Minor
Abstract: “The antihypertensive effect of galegine was investigated” or The hypotensive effect of galegine was investigated”
Abstract: “Meanwhile a positive control group”
Abstract: “intraperitoneal injection of galegine”
Abstract: “at the doses of 2.5, 5, and 10 mg/kg”
Abstract: “was the main effective compound”
Introduction: myocardial infarction, heart and kidney failure
All the wrong express have been corrected.
The 3D model of galegine is pixelized and should have a better resolution
The 3D model of galegine is obtained from Chemdraw, which was a better resolution.

Round 2
Reviewer 1 Report
The manuscript was revised based on the comments by this reviewer, and improved. This reviewer now recommends publishing it in the present form.
Author Response
Dear Reviewer;
English language and style were checked for times. Thank you for your patience.
Reviewer 2 Report
In this revised version, like in the previous one, the hypotensive effect of galegine does not appear in the Figure 3. This should be corrected accordingly (as I recommended, See figure 5 of
https://doi.org/10.3390/ijms22105106, which clearly shows hypertensive or hypotensive effects).
Author Response
the hypotensive effect of galegine does not appear in the Figure 4. This should be corrected accordingly (as I recommended, See figure 5 of
https://doi.org/10.3390/ijms22105106, which clearly shows hypertensive or hypotensive effects).
Yes, we have added Figure 5 to indicate the hypotensive effect of galegine. Figure 5 A: Representative raw traces of rat arterial blood pressure (ABP) following dimaprit or galegine infusion (blue arrow with doses used) compared to the control group. As expected, protocol assessment molecules dimaprit (5 mg.kg-1) and galegine(2.5 mg.kg-1, 5 mg.kg-1, 10 mg.kg-1) induced ABP, respectively. Figure 5 B : Box and whisker plots of ΔMABP data in the control, dimaprit and galegine group following i.p. injections, 5 mg.kg-1 for dimaprit and 2.5 mg.kg-1, 5 mg.kg-1, 10mg.kg-1 for galegine, respectively.